# Transcriptomic and Metabolomic Analyses Provide Insights into the Pathogenic Mechanism of the Rice False Smut Pathogen *Ustilaginoidea virens*

**DOI:** 10.3390/ijms241310805

**Published:** 2023-06-28

**Authors:** Rongtao Fu, Jian Wang, Cheng Chen, Yao Liu, Liyu Zhao, Daihua Lu

**Affiliations:** 1Institute of Plant Protection, Sichuan Academy of Agricultural Science, 20# Jingjusi Road, Chengdu 610066, China; furongtao@126.com (R.F.);; 2Key Laboratory of Integrated Pest Management on Crops in Southwest, Ministry of Agriculture, Chengdu 610066, China; 3Environment-Friendly Crop Germplasm Innovation and Genetic Improvement Key Laboratory of Sichuan Province, Crop Research Institute, Sichuan Academy of Agricultural Science, Chengdu 610066, China

**Keywords:** *Ustilaginoidea virens*, transcriptome, metabolome, autophagy-related gene, amino acids

## Abstract

Rice false smut, caused by the fungal pathogen *Ustilaginoidea virens*, is a worldwide rice fungal disease. However, the molecular mechanism of the pathogenicity of the fungus *U. virens* remains unclear. To understand the molecular mechanism of pathogenesis of the fungus *U. virens*, we performed an integrated analysis of the transcriptome and metabolome of strongly (S) and weakly (W) virulent strains both before and after the infection of panicles. A total of 7932 differential expressed genes (DEGs) were identified using transcriptome analysis. Gene ontology (GO) and metabolic pathway enrichment analysis indicated that amino acid metabolism, autophagy-yeast, MAPK signaling pathway-yeast, and starch and sucrose metabolism were closely related to the pathogenicity of *U. virens*. Genes related to pathogenicity were significantly upregulated in the strongly virulent strain, and were *ATG*, *MAPK*, *STE*, *TPS*, and *NTH* genes. However, genes involved in the negative regulation of pathogenesis were significantly downregulated and contained TOR kinase, *TORC1*, and autophagy-related protein genes. Metabolome analysis identified 698 differentially accumulated metabolites (DAMs), including 13 categories of organic acids and derivatives, lipids and lipid-like molecules, organoheterocyclic compounds. The significantly enriched pathways of DAMs mainly included amino acids and carbohydrates, and they accumulated after infection by the S strain. To understand the relevance of DEGs and DAMs in the pathogenicity of *U. virens*, transcriptomic and metabolomic data were integrated and analyzed. These results further confirmed that the pathogenesis of *U. virens* was regulated by DEGs and DAMs related to these four pathways, involving arginine and proline metabolism, lysine biosynthesis, alanine, aspartate and glutamate metabolism, and starch and sugar metabolism. Therefore, we speculate that the pathogenicity of *U. virens* is closely related to the accumulation of amino acids and carbohydrates, and to the changes in the expression of related genes.

## 1. Introduction

Rice false smut (RFS), caused by the fungus *Ustilaginoidea virens*, which infects rice panicles, is one of the main fungal diseases in the rice-growing regions worldwide [1]. The main symptom of this disease is the formation of rice false smut balls, which are several times larger than the grain. The fungus *U. virens* must absorb nutrients from rice to complete its vegetative growth and the formation of smut balls. This hinders grain filling, but also increases the sterility of grains near the smut ball. Thus, rice panicle nutrition deficiency and thousand seed weight decrease seriously threaten rice yield and quality [2]. The fungus produces mycotoxins, such as ustiloxins, water-soluble cyclic peptides, and ustilaginoidins, lipid-soluble naphthopyranones [3,4]. Mycotoxins inhibit the radicle elongation of seeds, but also have toxic effects on humans and animals [5].

In recent years, the severity and area of occurrence of RFS have increased due to the influence of rice varieties, cultivation conditions, and climate in China [6]. According to statistics, in the past 10 years, the average annual occurrence area of RFS has been more than 3 million hm^2^, and the average annual chemical control area has been 6.9 million hm^2^, accounting for 10% and 23% of the rice planting area, respectively [7]. At present, fungicide application is the most common and effective control method for RFS at the rice booting stage. The extensive use of chemical fungicides aggravates environmental pollution, and leads to the generation of fungicide-resistant strains [8]. Therefore, safe and accurate prevention and control of RFS can ensure rice yield and quality.

With the publication of the genome sequence of *U. virens*, the functional genes of *U. virens* have also been studied rapidly [9]. Lv et al. [10] reported that the fungus-specific transcription factor gene *UvPRO1* was involved in regulating the sporulation, stress response, and pathogenicity of *U. virens*. The *UvHOG1* gene of mitogen-activated protein kinase (MAPK) is involved in the regulation of the stress response, mycelial growth, and secondary metabolism of *U. virens* [11]. The autophagy-related gene *UvAtg8* is associated with mycelial growth, stress response, spore production, and the pathogenicity mechanism of *U. virens* [12]. The homebox transcription factor *UvHOX2* regulates chlamydospore formation and conidia production in *U. virens*, and the pathogenicity of *U. virens* is lost after gene deletion [13]. *Ustilaginoidea virens* causes RFS via a complicated process in specific infected rice-flower organs, which are involved in many metabolic pathways and genes [14,15] Therefore, it is necessary to systematically and deeply study the pathogenic mechanism of the pathogens.

With the development of omics technology, multiomics analysis has been widely used in the analysis of complex biological mechanisms in plants, among which transcriptomics and metabolomics are relatively mature, in-depth, and simple techniques [16,17]. In combined transcriptome and metabolome analysis, correlation analysis between the differentially expressed genes (DEGs) in the transcriptome and the differentially accumulated metabolites (DAMs) detected in the metabolome can be analyzed to determine the internal changes in organisms from the two levels of cause and result, showing the lock and key pathways of the changes in genes and metabolites, and building important regulatory networks to reveal its internal law [18]. Transcriptome and metabolome analyses have been used to study the stress and growth of pathogenic fungi, such as *Rhizopus oryzae* [19], *Agaricus bisporus* [20], and *Candida albicans* [21].

The uniomics transcriptome has been used to study biological processes related to interactions between compatible rice and *U. virens* [22] and the differential expression profiling of rice panicles’ response to *U. virens* mycotoxins [23]. In this study, we performed a multiomics transcriptome and metabolome analysis of strongly and weakly virulent strains before and after infection to reveal the pathogenic mechanism of *U. virens*. These results showed that metabolic pathways involved in arginine and proline metabolism, lysine biosynthesis, alanine, aspartate and glutamate metabolism, and starch and sugar metabolism, were closely related to the pathogenicity of *U. virens*. In addition, these genes play an important role in the pathogenicity of the pathogen, involving *ATG*, *MAPK*, *STE*, *TPS*, *NTH*, TOR kinase, *TORC1* and autophagy-related protein genes.

## 2. Results

### 2.1. Evaluation of U. virens Pathogenicity

In the pathogenicity identification of strains PXD25 and GY900 over 3 years, we found that PXD25 had a stronger pathogenicity than GY900 (Figure 1A). The pathogenicity rate of PXD25 was greater than 60%, while that of GY900 was less than 10% (Figure 1B). Therefore, PXD25 was referred to as a strongly virulent strain (S), and GY900 was known as a weakly virulent strain (W) [24].

### 2.2. Transcriptome Test Results and Analysis

#### 2.2.1. Summary of Transcriptome Data

To identify the pathogenic mechanism of *U. virens*, transcriptome sequencing was performed before and after infection of the strongly and weakly pathogenic strains. A total of 45,750,132 clean reads and 86.59 Gb clean bases were generated in 12 samples (Appendix A). Compared with the *U. virens* reference genome sequence, the efficiency of each sample was mapped at >95.92%. The error rate, Q20, Q30, and GC content values were less than 0.03%, 97.49–97.99%, 93.31–94.70% and 56.13–57.61%, respectively (Appendix A). These results indicate that the sequencing quality is high and suitable for further transcriptome analysis.

Pairwise comparisons of identified DEGs were performed before and after infection with the S and W strains. The |log_2_ fold-change| ≥ 1 and *p*-value < 0.05 were used as thresholds for DEG identification. A total of 6708 DEGs (3265 upregulated, 3443 downregulated) and 6521 DEGs (3224 upregulated, 3297 downregulated) were identified in S1 vs. S0 and W1 vs. W0, respectively (Figure 2A). A Venn diagram analysis showed DEGs that were common among strains with a difference in virulence, or specific to either strain in response to infected rice panicles (Figure 2B). Among them, 5297 were identified as DEGs common to S1 vs. S0 and W1 vs. W0; 1411 were categorized as DEGs only in S1 vs. S0, and 1224 were identified as DEGs only in W1 vs. W0 (Figure 2C).

#### 2.2.2. Enrichment Analysis of Functional Genes

To further investigate the functional classifications and biological functions of genes in strains PXD25 (S) and GY900 (W) after rice panicles’ infection, the gene ontology (GO) and pathway enrichment analyses of DEGs were performed. These data showed that the enriched GO terms were mainly related to molecular function, binding, catalytic activity, protein binding, ATP binding, organonitrogen compound metabolic process, oxidation reduction process, protein metabolic process, and protein binding (Figure 3A,B). KEGG enrichment analysis showed that 110 pathways were annotated, of which 28 pathways were significantly enriched in S1 vs. S0 and W1 vs. W0. Among them, the top 20 pathways are shown in Figure 3C,D. In S1 vs. S0 and W1 vs. W0, the primary pathways were autophagy-yeast, MAPK signaling pathway-yeast, lysine biosynthesis, arginine and proline metabolism, alanine, aspartate and glutamate metabolism, purine metabolism, TCA cycle, cysteine and methionine metabolism, and starch and sucrose metabolism.

#### 2.2.3. Analysis of Pathogenicity-Related DEGs

(1) Analysis of autophagy-related DEGs Previous studies have reported that autophagy plays an important role in the growth development and pathogenicity of plant pathogens [25]. The target of rapamycin (TOR) kinase plays a vital role in autophagy. It is a negative regulator that determines whether autophagy occurs [26]. In addition, some autophagy-related genes (*ATG*) and other autophagy-related factors induce autophagy. In this study, DEG analysis showed that the expression levels of 23 genes related to autophagy were changed in strains S and W after rice infection (Figure 4A and Appendix A). Among them, eight genes were downregulated in both S and W, namely three negative regulator genes (TOR kinase genes, *TORC1*, and *TapA*), two autophagy-related protein genes (*UV8b_02065* and *UV8b_03428*), two serine/threonine-protein kinase genes (*UV8b_04169* and *UV8b_01950*), and one phosphoinositide 3-kinase regulatory subunit 4 (*UV8b_00898*); 15 genes were upregulated, including five ATG genes (*ATG1*, *ATG4*, *ATG5*, *ATG7* and *ATG13*), one protein kinase SNF1 gene (*UV8b_04520*), one vesicle fusion factor NSFI (*UV8b_00760*), and one phosphoinositide 3-kinase (*PI3K*). Therefore, we speculated that *U. virens* can activate autophagy by inhibiting the expression of the TOR kinase gene, and activating the expression of *ATG* genes and other autophagy-related protein genes to improve the level of autophagy in response to the infection and pathogenicity of rice.

(2) Analysis of the *MAPK* signaling pathway DEGs Signal transduction genes of plant pathogenic fungi mainly have three categories, namely mitogen-activated protein kinase (MAPK) gene, cyclic AMP gene, and G protein gene, which coordinate the growth, development and pathogenicity of pathogens [27,28]. Here, the enrichment analysis of the signal transduction pathways of DEGs showed that MAPK was the main signal pathway after infection in strains S and W. A total of 12 genes were identified, with significant changes in expression (Figure 4B and Appendix A). Of these, four *MAPK* genes, two *STE* genes, two transcription factor genes, and one Rho GTPase activator (Sac7) gene were significantly upregulated in strains S and W after infection. These genes are closely involved in signal transduction. Therefore, our results indicate that MAPK is the main signaling pathway after *U. virens* infection in rice panicles.

(3) Analysis of amino acid metabolism DEGs Many studies have reported that amino acid metabolism is essential for the vegetative growth, infection cycle, and pathogenicity of plant pathogens [29,30]. In this study, we analyzed the DEGs in four amino acid metabolic pathways, namely arginine biosynthesis, lysine biosynthesis, arginine and proline metabolism, and alanine, aspartate and glutamate metabolism. A comparison of the expression levels of DEGs in these enrichment pathways indicated that many genes were upregulated in the strongly virulent strain but downregulated or inhibited in the weakly virulent strain (Figure 4C and Appendix A). Among them, multiple genes enriched in arginine and lysine biosynthesis were greatly upregulated in strain S after the infection of rice panicles. In contrast, these genes were not significantly regulated or suppressed in strain W. Taken together, our finding that amino acid metabolism-related genes showed opposite expression patterns between S and W after infection indicated that these genes played vital roles in *U. virens* pathogenicity. Additionally, the amino acid metabolism of *U. virens* may play an important role in the pathogenesis of *U. virens*.

(4) Analysis of starch and sucrose metabolism DEGs Starch and sucrose metabolism play an important role in the development and pathogenicity of plant fungal pathogens [31]. To identify potential regulatory genes closely associated with starch and sucrose metabolism, we identified DEGs by comparing the expression changes between strains S and W. Of the genes involved in starch and sucrose metabolism that were expressed after the infection of rice panicles, more were expressed at a higher level in strain S than in strain W; the difference between the two strains is shown in a heatmap (Figure 4D and Appendix A). Previous studies have found that trehalose is closely related to the pathogenicity of plant pathogens [32]. Here, we found four genes involved in trehalose metabolism, namely three trehalose-6-phosphate synthase (TPS) genes and one neutral trehalase (NTH) gene. The expression levels of these genes were upregulated in strains S and W, but the expression in strain S was significantly higher than in strain W. Therefore, the results showed that these DEGs involved in starch and sugar metabolism might play an essential role in the virulence of *U. virens*.

#### 2.2.4. RT-qPCR Validation of DEGs

To validate the reliability and authenticity of the DEG data from transcriptome analysis, the expression levels of 10 candidate genes were confirmed using RT-qPCR. Each assay was conducted in triplicate with Tubulin as the reference gene. The RT-qPCR results showed that the expression patterns of the candidate genes were consistent with the RNA-Seq analysis (Table 1). Therefore, the RT-qPCR results suggest that the transcriptome data are reliable.

### 2.3. Detection Results and Analysis of the Untargeted Metabolome

#### 2.3.1. Quality Control and Statistical Analysis of Untargeted Metabolome Data

A total of 19 samples were analyzed using metabolomics, comprising the treatment group (S1 and W1) and control group (S0 and W0), with four biological replicates for each treatment, and three quality control samples (QC). Qualitative and quantitative analyses of metabolites were performed using the UHPLC-QE-MS detection platform. Mass spectrometry of QC analysis showed that the total ion flow of metabolites had high overlap in the positive and negative ion modes (Figure 5A,B). As shown in the figures, the technology in this experiment had high reliability and overlap, demonstrating the reliability and authenticity of the data. The PCA analysis of the peaks obtained from all samples showed that the samples were closely clustered together in the positive and negative ion modes, indicating the good repeatability of the experiment (Figure 5C,D). Therefore, the metabolome data of this experiment are real and reliable, and can be used for further analysis.

#### 2.3.2. Qualitative and Quantitative Analyses of Metabolites

The metabolites in strongly and weakly virulent strains were detected qualitatively and quantitatively twice using the untargeted metabolome determination method. A total of 698 metabolites were detected in this experiment, including 186 organic acids and derivatives, 133 lipids and lipid-like molecules, 86 benzenoids, 81 organic oxygen compounds, 8 alkaloids and derivatives, 31 phenylpropanoids and polyketides, and 13 others (Table 2). In addition, the cluster heat map of the metabolites in the samples (Figure 6A) showed that there were significant differences in the metabolite content in the strongly virulent (S) and weakly virulent (W) strains before and after infection. Furthermore, the orthogonal partial least squares discriminant analysis (OPLS-DA) score diagram (Figure 6B) and the model validation diagram of OPLS-DA (Figure 6C) of all samples from each group indicated significant differences in S and W strains before and after infection.

#### 2.3.3. Differential Metabolite Analysis

Metabolites with VIP ≥ 1 and *p*-value < 0.05 were designated as DAMs. Consequently, 562 DAMs were detected. Among them, there were 452 DAMs (223 upregulated, 229 downregulated) in S1 vs. S0, and 392 DAMs (188 upregulated and 204 downregulated) in W1 vs. W0 (Figure 7A,B). The Venn diagram showed that there were 282 common DAMs in S1 vs. S0 and W1 vs. W0, 170 and 110 unique DAMs in the S1 vs. S0 and W1 vs. W0, respectively (Figure 7C). Spearman correlation analysis was used to analyze the correlation of the top 50 DAMs with the highest significant difference (Appendix A). The correlation heatmap showed that the absolute value of positive and negative correlation coefficients between DAMs ranged from 0.636 to 0.999, indicating that the DAMs with significant differences had significant changes in convergence or anisotropy.

To better understand the pathways involved in DAMs, KEGG enrichment analysis was performed for DAMs. Figure 7D,E shows the top 20 pathways enriched in the two combinations. A total of 82 metabolic pathways were enriched in S1 vs. S0, of which 10 metabolic pathways were significantly different, including the TCA cycle, arginine and proline metabolism, ABC transporters, pentose phosphate pathway, starch and sugar metabolism, biosynthesis of amino acids, glycine, serine and threonine metabolism, lysine biosynthesis, carbon metabolism, and alanine, aspartate, and glutamate metabolism (Appendix A). W1 vs. W0 involved a total of 79 metabolic channels, including six metabolic channels, namely arginine biosynthesis, arginine and proline metabolism, alanine, aspartate and glutamate metabolism, phenylalanine, tyrosine and tryptophan biosynthesis, starch and sugar metabolism, and lysine biosynthesis, with significant differences (Appendix A). It can be seen from the significantly accumulated metabolites that most of the accumulated metabolites increased in the strongly virulent strain (S1) after infection and were higher than those in the weakly virulent strain (W1) (Figure 7F). These metabolites mainly include amino acids, carbohydrates, and other metabolites (Figure 7F), indicating that these substances may play an important role in the pathogenesis of *U. virens*.

### 2.4. Combined Analysis of Transcription and Metabolome

#### 2.4.1. Correlation Analysis between DEGs and DAMs

To understand the relevance of DEGs and DAMs in the pathogenicity of *U. virens*, transcriptomic and metabolomic data were integrated and analyzed. Correlation analysis was performed for the genes and metabolites detected in each group based on the Pearson correlation coefficient. The nine-quadrant diagram (Appendix A) showed the differential multiples of the metabolites and genes with Pearson correlation coefficients greater than 0.8 in each differential group, and the differentially expressed patterns of genes and metabolites were consistent in quadrants three and seven, indicating that these metabolites were positively regulated by genes. In quadrants one and nine, the differentially expressed patterns of genes and metabolites were opposite, and these metabolites were negatively regulated by genes. For example, in S1 vs. S0, 3223 genes positively regulated 363 metabolites and 3207 genes negatively regulated 363 metabolites. W1 vs. W0 had 2812 genes that positively regulated 331 metabolites and 2810 genes that negatively regulated 331 metabolites.

The common enrichment pathways of DAMs and DEGs were obtained by KEGG enrichment analysis. As shown in Appendix A, 68 metabolic pathways were co-enriched in S1 vs. S0, and 64 metabolic pathways were co-enriched in W1 vs. W0. The KEGG enrichment analysis bubble diagram (Appendix A) showed that the DEGs and DAMs of strongly and weakly virulent strains were significantly enriched in arginine and proline metabolism, lysine biosynthesis, alanine, aspartate and glutamate metabolism, and starch and sugar metabolism. It was further confirmed that the pathogenesis of *U. virens* was regulated by DEGs and DAMs related to these four pathways.

#### 2.4.2. Correlation Analysis of DEGs and DAMs in Significantly Enriched Metabolic Pathways

To further investigate the correlation of DEGs and DAMs in “arginine and proline metabolism, lysine biosynthesis, alanine, aspartate and glutamate metabolism, and starch and sugar metabolism” in the infection mechanism of *U. virens*, we compared S1 and S0 and drew a network diagram to reveal the relationship between genes and metabolites. This experiment selected DAMs and DEGs with a correlation greater than 0.8 for the correlation network map. In arginine and proline metabolism, nine metabolites displayed a strong correlation with 17 genes (Figure 8A and Appendix A). The metabolites contained l-arginine, 4-aminobutyric acid (GABA), s-adenosylmethionine, 4-acetamidobutanoate, sarcosine, n-(omega)-hydroxyarginine, glutamic, pyruvate, and d-proline. Nine related enzymes, namely pyrroline-5-carboxylate reductase [EC: 1.5.1.2], gamma-glutamyl phosphate reductase [EC: 1.2.1.38], acetamidase [EC: 1.7.1.12], general amidase [EC: 3.2.1.45], s-adenosylmethionine decarboxylase proenzyme [EC: 1.1.1.170], histone acetyltransferase [EC: 2.3.1.48], delta 1-pyrroline-5-carboxylate dehydrogenase [EC: 1.2.1.88], aspartate aminotransferase [EC: 2.6.1.1], and ornithine aminotransferase [EC: 2.6.1.13], were identified. Among them, l-arginine and s-adenosylmethionine had a correlation higher than 0.99 with the following genes: *UV8b_00446*, *UV8b_06678*, *UV8b_01984*, *UV8b_02871*, *UV8b_04468*, and *UV8b_07070*. Pyruvate was negatively regulated by general amidase, s-adenosylmethionine decarboxylase proenzyme, pyrroline-5-carboxylate reductase, aspartate aminotransferase, delta 1-pyrroline-5-carboxylate dehydrogenase, and gamma-glutamyl phosphate reductase. Seven metabolites were negatively regulated by the *UV8b_06541* gene.

In lysine biosynthesis, four metabolites showed a high relationship with 12 genes (Figure 8B and Appendix A). The metabolites included l-lysine, l-saccharopine, oxoadipic acid, and homocitrate. These genes encoded seven enzymes, namely homocitrate synthase [EC: 2.3.3.14], homoserine dehydrogenase [EC: 1.1.1.3], aspartate-semialdehyde dehydrogenase [EC: 1.2.1.11], aminoadipate reductase enzyme [EC: 1.2.1.31], saccharopine dehydrogenase [EC: 1.5.1.7], aconitate hydratase [EC: 4.2.1.3], and aspartokinase [EC: 2.7.2.4]. L-lysine had a strong correlation with the genes *UV8b_06216*, *UV8b_07007*, *UV8b_05599*, *UV8b_00056*, and *UV8b_02651*, and these genes positively regulated l-lysine.

In alanine, aspartate, and glutamate metabolism, six metabolites were high correlation with 17 genes (Figure 8C and Appendix A). Six DAMs were involved: fumarate, pyruvate, succinic semialdehyde, glutamic acid, 4-aminobutyric acid (GABA), and citrate. The enzymes involved included fructose-6-phosphate amidotransferase [EC: 2.6.1.16], delta 1-pyrroline-5-carboxylate dehydrogenase [EC: 1.2.1.88], succinate semialdehyde dehydrogenase [EC: 1.2.1.16], amidophosphoribosyltransferase [EC: 2.4.2.14], glutamine synthetase [EC: 1.4.1.13], asparagine synthetase [EC: 6.3.5.4], glutamate decarboxylase [EC: 4.1.1.15], 4-aminobutyrate aminotransferase [EC: 2.6.1.96], and NAD+ dependent glutamate dehydrogenase [EC: 1.4.1.2]. Among them, glutamic acid had a high correlation with the genes *UV8b_06120*, *UV8b_01796*, *UV8b_07879*, *UV8b_04621*, *UV8b_02433*, *UV8b_02378*, and *UV8b_00289*, and positively regulated glutamic acid.

In starch and sugar metabolism, four metabolites were involved: glucose-6-phosphate, trehalose, sucrose, and d-fructose (Figure 8D and Appendix A). The enzymes involved were neutral trehalase [EC: 3.2.1.28], trehalose phosphate synthase [EC: 2.4.1.347], maltase MLT3 [EC: 3.2.1.20], glucoamylase GMY2 [EC: 3.2.1.3], glycosyltransferase family 20 protein [EC: 2.4.1.25], beta-1,3-glucanase [EC: 3.2.1.6], alpha-glucosidase [EC: 3.2.1.20], hexokinase [EC: 2.7.1.1], and glucose-6-phosphate isomerase [EC: 5.3.1.9].Using correlation analysis, we found that the genes *UV8b_06341*, *UV8b_05798*, *UV8b_05883*, *UV8b_04918*, and *UV8b_04913* had a strong correlation with glucose-6-phosphate (∣PPC∣ > 0.995), these genes positively regulated glucose-6-phosphate. Trehalose was positively regulated by genes: *UV8b_07751*, *UV8b_05798*, and *UV8b_00809*, but negatively regulated by UV8b_04523. Three genes showed a high correlation with sucrose, namely *UV8b_01012*, *UV8b_04240*, and *UV8b_05361*, all of which were positively regulated.

## 3. Discussion

RFS is a major fungal disease in the rice-growing areas of the world, seriously threatening safe rice production. RFS not only reduces rice yield and quality, but also produces toxins that can cause harm to human and animal health. At present, most studies have focused on morphology [33], infection processes [15], genetic diversity [34], and mycotoxin [4,5] in the causal agent of RFS, *U. virens*. An in-depth analysis of the pathogenic mechanism of *U. virens* is of great guiding significance for the development of new fungicides and breeding for disease resistance; however, little is known concerning these topics. In view of this, the aim of this study was to investigate the pathogenesis of *U. virens* based on two omics transcriptomes and metabolomes.

Combined transcriptome and metabolome analysis has been used to explore a series of important problems, such as plant response to the molecular mechanism of pathogen infection [16] and the growth and development mechanisms of pathogenic fungi [19]. In this study, transcriptome and metabolome analyses were used for the first time to study the changes in genes and metabolites in pathogen *U. virens* after infection of the host to explore the molecular mechanism of pathogenesis in the pathogen. First, comparative transcriptome analyses showed that the expression levels of 7932 genes were significantly changed in the strongly and weakly virulent strains after infection. GO term analysis of the DEGs showed that molecular function and binding were the most enriched in the molecular function category. Based on KEGG analysis, most DEGs were significantly enriched in the following pathways: autophagy-yeast, MAPK signaling pathway-yeast, lysine biosynthesis, arginine and proline metabolism, alanine, aspartate and glutamate metabolism, purine metabolism, TCA cycle, cysteine and methionine metabolism, and starch and sucrose metabolism. Previous studies have reported that metabolic pathways, such as autophagy, MAPK signaling pathway, amino acid metabolism, and TCA cycle, are closely related to the vegetative growth, development, and pathogenicity of plant pathogens [29,35,36,37].

Autophagy is a highly conserved degradation pathway of cytoplasmic contents in eukaryotes, by which some long-lived proteins and damaged organelles are degraded to maintain cell homeostasis and help them survive adversity, such as starvation or environmental stress [38,39]. In recent decades, autophagy has been found to play an important role in the pathogenicity of plant pathogenic fungi. The autophagy process is complex and regulated by autophagy negative regulatory factors (TOR kinases), and many autophagy proteins are involved in each step, such as ATG, SNF1 kinase, and PI3K [26,38,40]. In this study, three negative regulator genes (TOR kinase genes, TORC1, and TapA) were significantly downregulated, while five *ATG* genes, one SNF1 kinase gene and *PI3K* were significantly upregulated in the S strain after infection. TOR is a major regulator of autophagy and determines whether autophagy occurs [26]. Previous studies have reported that the autophagy-related gene *UvAtg8* is associated with the mycelial growth, stress response, spore production and pathogenicity of *U. virens* [12]. Thus far, several ATG have been found to be necessary for the pathogenicity of *Magnaporthe grisea* [35,39,40]. Taken together, these findings indicate that autophagy regulatory factors and *ATG* are closely related to the pathogenicity of *U. virens*.

In recent years, the MAPK signaling pathway has been shown to play an important role in the growth, development, propagation, and pathogenicity of various plant pathogenic fungi [41,42]. Currently, three types of MAPK cascade kinases have been identified in plant pathogenic fungi, namely MAPK, MAPKK and MAPKKK, and these cascade kinases play an important role in the mycelial infection, stress response, and virulence of plant pathogenic fungi [43]. Consistent with previous findings, this study showed that MAPK was the main signal pathway in *U. virens* after infection. We found that two *MAPK*, one *MAPKK*, one *MAPKKK*, and other regulator factor genes were significantly upregulated in strain S after infection. Previous studies have shown that the strains that knock out *MAPKK* and *MAPKKK* in *M. grisea* lose their ability to infect [42]. In *Colletotrichum higginsianum*, mutation of the *MAPK* gene significantly reduces infectivity and pathogenicity [43]. Therefore, the results indicate that the MAPK cascade signaling pathway and related kinases play an important role in the infection and pathogenesis of *U. virens*.

Many studies have shown that amino acid metabolism plays an important role in the infection cycle and pathogenicity of pathogens [29,37,44]. It has been reported that many amino acid metabolisms, such as “arginine biosynthesis, lysine biosynthesis, arginine and proline metabolism, and “alanine, aspartate and glutamate metabolism”, are crucial to the pathogenicity of plant pathogens [29,45,46]. In this study, DEG analysis revealed that the expression levels of many genes involved in these four metabolic pathways were significantly upregulated in the strongly virulent strain (S), and downregulated or inhibited in the weakly virulent strain (W). Metabolome analysis found that metabolites involved in four metabolic pathways, namely l-arginine, l-lysine, glutamic, d-proline, and threonine, were also significantly upregulated in strain S after infection. Previous studies have shown that these metabolites are closely related to the pathogenicity of plant pathogens, such as *M. oryzae* [29,46], *Botrytis cinerea* [45], and *Fusarium graminearum* [47]. Taken together, our results suggest that the genes and metabolites involved in amino acid metabolism are closely related to the virulence of *U. virens*.

Starch and sugar metabolism provide an important carbon source for mycelial infection by plant pathogens, which are closely related to pathogenicity [31,47]. In this study, the expression of many genes involved in the metabolic pathways of starch and sucrose such as *TPS*, alpha-glucosidase, and beta-glucosidase genes, was significantly up-regulated in strains S after infection. Previous studies have found that knockout of the *TPS* gene in *M. oryzae* resulted in the loss of the trehalose synthesis ability of the mutant, decreased infection and growth ability, and severely weakened pathogenicity [32]. In this experiment, metabolome analysis found that metabolites involved in starch and sugar metabolism were significantly accumulated in a strongly virulence strain (S) after infection, especially trehalose, consistent with the results of previous studies. Studies have shown that trehalose can be a virulence factor that determines the pathogenicity of pathogens [48,49]. Collectively, DEGs and DAMs involved in starch and sugar metabolism might play an essential role in the virulence of *U. virens*. In addition, trehalose may play an important role in the infection process of *U. virens*.

Many studies have shown that the combined analysis of transcriptomic and metabolomic data can yield functional insights into different biological processes [16,18]. In fungi, transcriptomics and metabolomics have been used to study stress and growth; for example, *Rhizopus oryzae* resists oxidative stress by upregulating oxidation pathways and inducing some antioxidant enzymes [19]. In this study, to obtain an in-depth analysis of changes in the pathogen *U. virens* after infection, combined analysis of the transcriptome and metabolome was performed. Three amino acid metabolism pathways (arginine and proline metabolism, lysine biosynthesis, and “alanine, aspartate and glutamate metabolism”) and “starch and sugar metabolism” were significantly co-enriched in combined analysis. Therefore, we speculate that these three amino acid metabolism pathways and “starch and sugar metabolism” may be the key pathways of the pathogenicity of *U. virens*. In addition, in the study results, we found that the metabolites in these four metabolic pathways had a strong correlation with their regulatory genes, especially arginine, lysine, and glutamic, which showed positive or negative regulation.

In conclusion, this study revealed the in-depth pathogenic mechanism of the fungus *U. virens* through transcriptomic and metabolomic analyses. Comparative transcriptome analysis showed that many metabolic pathways, including amino acid metabolism, autophagy-yeast, MAPK signaling pathway-yeast, and starch and sucrose metabolism, were activated in *U. virens* after infection. In strongly virulent strains, positive regulatory genes related to pathogenicity were significantly upregulated, including *ATG*, *MAPK*, *STE*, *TPS,* and *NTH* genes. However, genes involved in negative regulation were significantly downregulated and contained TOR kinase, *TORC1*, and autophagy-related protein genes. Metabolome analysis showed that the significantly enriched pathways of DMAs mainly included amino acids and carbohydrates and largely accumulated in the strongly virulent strain after infection. These results suggest that these metabolic pathways, genes, and metabolites are closely related to the pathogenicity of *U. virens*. Moreover, integrated transcriptome and metabolome analysis further indicated that these pathogenic factors were closely related to the pathogenicity of *U. virens*. Further research is needed to determine the mechanism of action of these pathogenicity factors in pathogenic *U. virens*. These findings increase our understanding of the molecular mechanisms of the pathogenicity of *U. virens*.

## 4. Materials and Methods

### 4.1. Materials and Fungal Inoculation

The pathogen *U. virens* was isolated from the yellow false smut balls. The isolation methods and culture conditions of the pathogen were as previously reported by Ashizawa et al. [15]. From 2019 to 2021, the pathogenicity of the GY900 and PXD25 strains was measured. As a result (Figure 1), GY900 was identified as a weakly virulent strain (S), and PXD25 was identified as a strongly virulent strain (W). The isolates GY900 and PXD25 were grown on potato sucrose agar (PSA) medium and mycelium disks were placed in potato sucrose (PS) fluid medium. The cultures were incubated at 27 °C on a shaker at 120 rpm for 14 days. Mycelia and conidia were collected for inoculation.

*Oryza sativa* L. spp. *indica* cultivar ‘93–11’ was used in this experiment. The rice plants were grown in controlled greenhouses with the temperatures ranging from 20 °C at night to a maximum of 36 °C during the day. The inoculation protocols described by Ashizawa et al. [15] and Fu et al. [23] were used with minor modifications. At the seventh to eighth stage of panicle development, the conidial suspension with a concentration of 3 × 10^6^ mL^−1^ (GY900 and PXD25 strains) was injected into each rice panicle. The controls were injected with PS fluid medium for all experiments. After inoculation, all the rice plants maintained at 25/30 °C (night/day), covered with an inner solar-shade screen, and automatically sprayed water every 4 h for 20 min to maintain the environment at relative humidity (RH) >85% for 4 days. Samples of pathogens before and after infection with mycelium were collected, immediately frozen in liquid nitrogen, and kept at −80 °C for later use. The samples of strongly virulent strains were named S0 and S1 and weakly virulent strains were named W0 and W1 for before and after infection, respectively.

### 4.2. Transcriptome Sequencing and Analysis

The total RNA of the biological samples was isolated using Trizol (Aidlab Biotechnologies, Beijing, China) following the manufacturer’s protocol. There were three biological replicates per sample group. The quality and quantity of the extracted RNA were detected using 1% agarose gel and an Agilent 2100 Bioanalyzer (Agilent Technologies, Santa Clara, CA, USA). The obtained high-quality RNA was sent to the company (Allwegene Co., Ltd., Nanjing, China) for library construction and sequencing, using an HiSeqTM2500 platform (pair-end 150 bp).

For sequence data analysis, low-quality reads were filtered out of the original data under the default parameters to obtain clean reads. The clean reads were mapped onto the reference genome sequence of *U. virens* using HISAT2 [50] to obtain the sample genes. No more than two mismatches were allowed in the alignment for each read. The gene expression level was counted using HTSeq software (v 0.5.4) and normalized using the fragments per kilobase per million fragments (FPKM) [51]. DEseq2 was used to analyze the DEGs between the two groups. DEGs were identified by comparing gene expression levels between the before and after infection groups of the S and W strains with a *p*-value < 0.05 and |log2foldchange| ≥ 1 [52]. The DEGs were then subjected to enrichment analysis of Gene Ontology (GO) and Kyoto Encyclopedia of Genes and Genomes (KEGG). A q-value ≤ 0.05 was considered as the threshold for significant enrichment of GO and KEGG pathways in DEGs.

### 4.3. Untargeted Metabolome Detection and Analysis

To identify differences in metabolites before (mycelium) and after infection of strongly and weakly virulent *U. virens*, we selected 16 samples (four in each group, four groups) for metabolomic analysis. Untargeted metabolomics was performed using an Ultra-high performance liquid chromatography Q exactive mass spectrometer (UHPLC-QE-MS) following the previously reported methods with minor modifications [53]. Briefly, the samples were ground to powder under liquid nitrogen. Then, 100 mg powder was weighed and dissolved in 1.0 mL of methanol/acetonitrile/water solution (2:2:1, *v*/*v*), vortexed, ultrasonicated at a low temperature for 30 min, and left to stand at −20 °C for 10 min. Following centrifugation at 4 °C for 20 min at 14,000× *g*, the supernatant was dried with a vacuum. The dry substance was redissolved in aqueous acetonitrile, and the supernatant was obtained by centrifugation for analysis using UHPLC. The samples were separated by Vanquish LC UHPLC and then analyzed by mass spectrometry using a Q Exactive series mass spectrometer (Thermo, Waltham, MA, USA). Positive and negative ion modes of electrospray ionization (ESI) were used for detection.

After the mass spectrum analysis data of metabolites in different samples were obtained by triple quadrupole mass spectrometry, the mass spectrum peaks of all substances were integrated, and the mass spectrum peaks of the same metabolites in different samples were calibrated. The data were then analyzed qualitatively and quantitatively. Unsupervised principal component analysis (PCA) was carried out using the statistical function prcomp in R (http://www.r-project.org/ accessed on 26 October 2022) to observe the overall distribution among samples and the stability of the whole analysis process [54,55]. The metabolites between different groups were distinguished by orthogonal partial least squares-discriminant analysis (OPLS-DA) and partial least squares-discriminant analysis (PLS-DA). Variable importance of projection (VIP) values obtained from the OPLS-DA model were applied to rank the overall contribution of each variable to group discrimination [55]. The threshold of significant difference was set as VIP ≥ 1 and *t*-test *p* < 0.05 to identify DAMs between the two groups. The KEGG and GO databases were then used to annotate the functions of the DAMs, and the metabolic pathways involving in the metabolites were obtained.

### 4.4. Combined Analysis of the Transcriptome and Metabolome

Based on the metabolome and transcriptome results, the common KEGG pathways of DEGs and DAMs were analyzed jointly. Correlation analysis of the detected genes and metabolites was performed using the cor program in the R package to calculate the Pearson correlation coefficients of genes and metabolites. The gene metabolites with a Pearson correlation coefficient > 0.8 and *p*-value < 0.05 in each group were screened as network maps to show the correlation between metabolites and genes [55].

### 4.5. Validation of qRT-PCR

To validate the DEG results, qRT-PCR was carried out using 10 DEGs, with tubulin as the internal reference. The primer sets used for qRT-PCR were designed according to the individual gene sequences (Table 1). The same RNA sample used for RNA-Seq was used for qRT-PCR. cDNA was produced by reverse transcription using the Fist-Strand cDNA Synthesis Kit (TransScript, Beijing, China) according to the manufacturer’s protocol. qRT-PCR was conducted using Takara SYBR Green (Takara, Dalian, China) in a one-step, real-time system. There were three biological replicates for each sample, and the reaction conditions were as follows: 95 °C for 1 min, followed by 40 cycles of 95 °C at 10 s and 60 °C for 30 s. The 2^−ΔΔCT^ method was used to calculate the relative transcription level of the genes [56].

### 4.6. Statistical Analysis

The data from this experiment were expressed as the means ± SD of three replicates for each treatment. Statistical analysis was performed using a one-way analysis of variance (ANOVA) and SPSS 20.0 (IBM, Armonk, NY, USA). A *p* < 0.05 indicated a statistically significant difference.

## Figures and Tables

**Figure 1 ijms-24-10805-f001:**
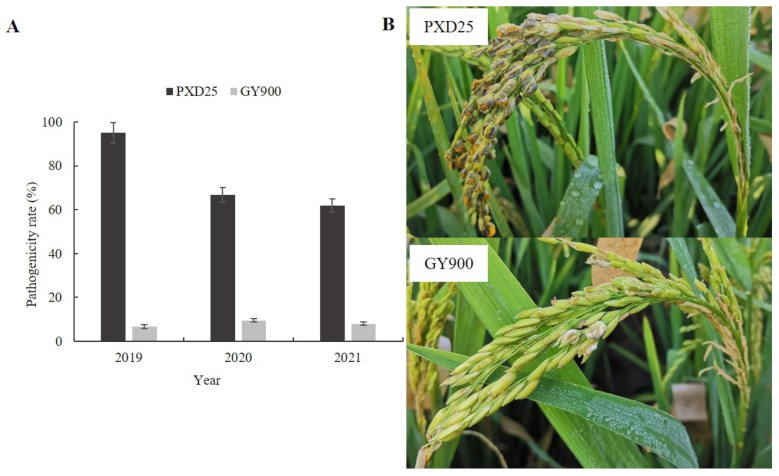
Evaluation of *Ustilaginoidea virens* pathogenicity. (**A**) Pathogenicity identification of strains PXD25 and GY900. (**B**) The disease symptoms of strains PXD25 and GY900 in the field.

**Figure 2 ijms-24-10805-f002:**
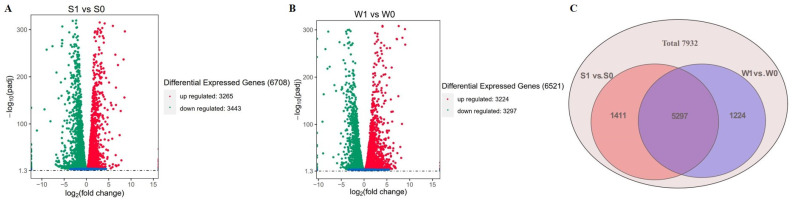
Analysis of differentially expressed genes (DEGs) in S1 vs. S0 and W1 vs. W0. (**A**) Volcano plot of DEGs in S1 vs. S0. (**B**) Volcano plot of DEGs in W1 vs. W0. (**C**) Venn diagram of DEGs.

**Figure 3 ijms-24-10805-f003:**
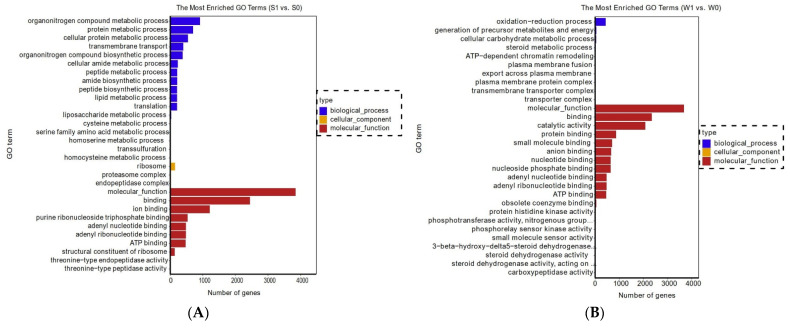
Gene ontology (GO) and KEGG functional classification. (**A**) Most enriched GO terms in S1 vs. S0. (**B**) Most enriched GO terms in W1 vs. W0; (**C**) top 20 pathways in S1 vs. S0; (**D**) top 20 pathways in W1 vs. W0.

**Figure 4 ijms-24-10805-f004:**
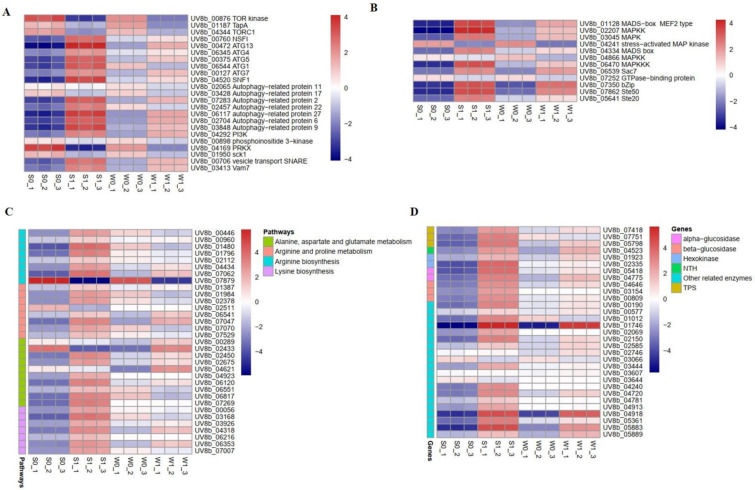
Expression pattern of differentially expressed genes (DEGs) in strains S and W after infection of rice panicles. (**A**) Analysis of autophagy-related DEGs. (**B**) Analysis of MAPK signaling pathway DEGs. (**C**) Analysis of amino acid metabolism DEGs. (**D**) Analysis of starch and sucrose metabolism DEGs.

**Figure 5 ijms-24-10805-f005:**
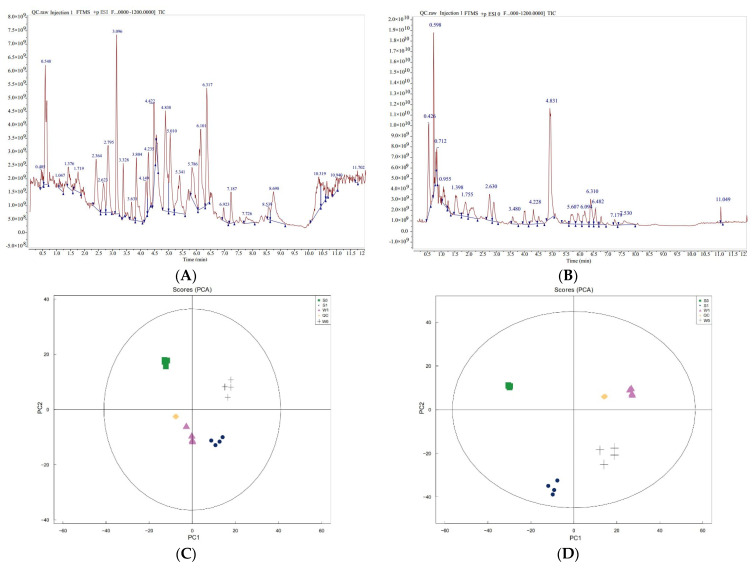
Metabolome data quality evaluation. (**A**) Positive ion chromatogram overlap spectrum of quality control samples. (**B**) Negative ion chromatogram overlap spectrum of quality control samples. (**C**) PCA analysis of total samples in positive ion mode. (**D**) PCA analysis of total samples in negative ion mode.

**Figure 6 ijms-24-10805-f006:**
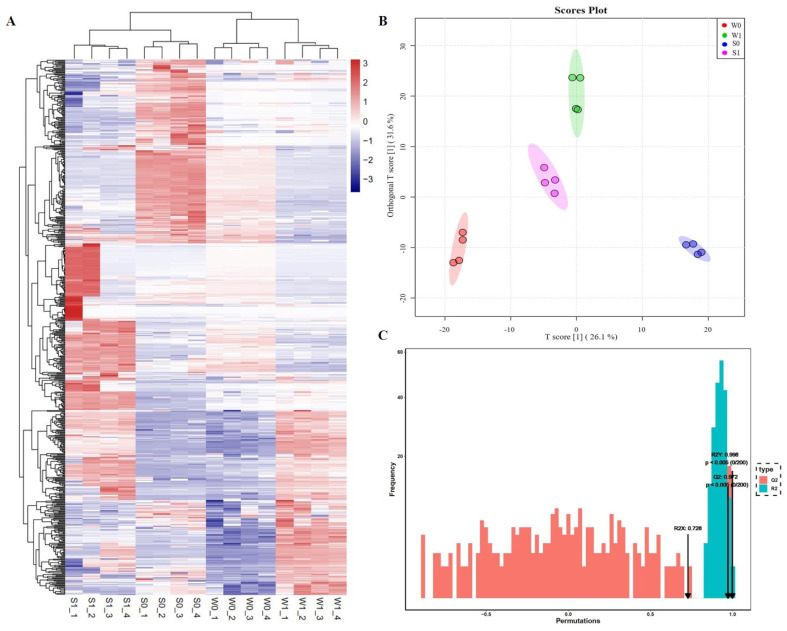
Qualitative and quantitative analysis of differentially accumulated metabolites (DMAs). (**A**) Hierarchical cluster analysis of heat maps of all DAMs in all samples. (**B**) OPLS-DA diagram of all samples. (**C**) OPLS-DA model validation diagrams for all samples.

**Figure 7 ijms-24-10805-f007:**
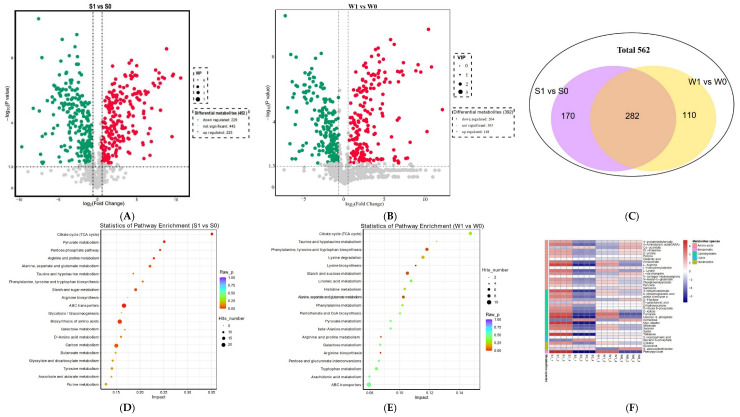
Differential metabolite analysis. (**A**) Volcano plot of differentially accumulated metabolites (DAMs) in S1 vs. S0. (**B**) Volcano plot of DAMs in W1 vs. W0. (**C**) Venn diagram of DAMs. (**D**) Top 20 pathways enriched in S1 vs. S0. (**E**) Top 20 pathways enriched in W1 vs. W0. (**F**) Heat maps of DAMs.

**Figure 8 ijms-24-10805-f008:**
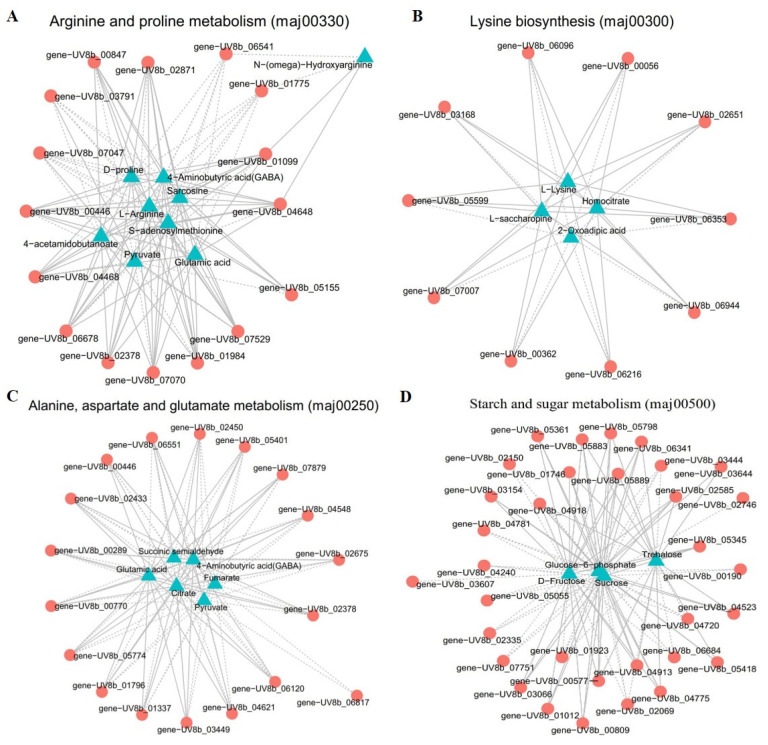
Correlation network analysis of DAMs and DEGs. (**A**) arginine and proline metabolism. (**B**) lysine biosynthesis. (**C**) alanine, aspartate, and glutamate metabolism. (**D**) starch and sugar metabolism.

**Table 1 ijms-24-10805-t001:** qRT-PCR primers used in this study to verify the RNA-Seq data.

Genes	Primers(5′–3′)	Log2FC (RNA-Seq)	Log2FC (RT-qPCR)
UV8b_04520	F-ATGGCTCCCCGAGGAGGCTTTG R-TCAGTCCGCCTCTGCCAGCTGCATGA	3.34251	2.93451
UV8b_06544	F-ATGGCGAGTCGTCAAGATGGAT R-TTATGCTGATCCATGCGATGGTA	2.95398	2.56875
UV8b_07283	F-ATGCCCAAGCGCCTACTCCGT R-CGGAGTCTGCTGCAGCCAAACCCTA	2.97542	2.28723
UV8b_07350	F-ATGGGGACTTCTCCTGCCACCGAGT R-TCATGAGAAGCGGCGCTGGACGCCT	3.32256	3.01291
UV8b_05641	F-ATGGACGGTCCGAGATCGTCCTTG R-GATGTTGCCTTCGTTGGACAGCA	2.41345	2.13214
UV8b_01480	F-ATGGCAGGCGGCATGGGTCCCC R-TTAGCAGTATGACTCGTTAGGC	3.90424	3.45192
UV8b_06216	F-ATGGCAAATGCAAACTCCATCTA R-TCAGGTAATCTGGGCAATAACC	1.63942	1.02376
UV8b_07418	F-ATGTCCAGTGACACCAGTATTAC R-AATTCTATGGTACCGAATTTGCCGT	2.56132	1.98124
UV8b_04169	F-ATGGCGGCAACGGTAGCCCAGTCG R-TCAAAACTCCTCGAAGTACCGG	−3.60581	−3.97812
UV8b_00876	F-ATGGCGCAACAACAGCAGGTC R-CCTTCCCTGACATATATGAGCA	−3.27163	−3.51234

**Table 2 ijms-24-10805-t002:** Types and quantities of metabolites.

Species of Metabolites	Number of Metabolites
Organic acids and derivatives	186
Lipids and lipid-like molecules	133
Organoheterocyclic compounds	100
Benzenoids	86
Organic oxygen compounds	81
Phenylpropanoids and polyketides	31
Nucleosides, nucleotides, and analogues	28
Organic nitrogen compounds	25
Other	13
Alkaloids and derivatives	8
Organosulfur compounds	2
Lignans, neolignans and related compounds	2
Homogeneous non-metal compounds	2
Hydrocarbon derivatives	1
Total	698

## Data Availability

The datasets presented in this study can be found in online repositories. The names of the repository/repositories and accession number(s) can be found below: https://www.ncbi.nlm.nih.gov/, PRJNA974935.

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
