# Peer review of "Transcriptomic and Metabolomic Analyses Provide Insights into the Pathogenic Mechanism of the Rice False Smut Pathogen Ustilaginoidea virens"

_ijms, 2023, doi:10.3390/ijms241310805_

Round 1

Reviewer 1 Report

Dear colleagues.
There are a few comments. Ones are in the attached file

Author Response

Dear reviewer

We would like to thank you for giving us the opportunity to revise our manuscript entitled “Transcriptomic and metabolomic analyses provide insights into the pathogenic mechanism of the rice false smut pathogen Ustilaginoidea virens”. We have been carefully revised in the revised manuscript and the every checking was marked in red.

We thank you for their careful read and thoughtful comments on previous manuscript. The point-by-point answers to the comments and suggestions were listed as below.

# Reviewer 1

Comments and Suggestions for Authors

Line 83.

In addition, genes related to pathogenicity play an important role in the pathogenicity of the pathogen, involving ATG, MAPK, Ste, TPS, NTH, TOR kinase

The construction of the phrase seems redundant

Answer: According to your comments, we have made corresponding modifications in revised manuscript.

Line 101-108.

Oryza sativa L. spp. indica cultivar ‘93-11’ was used in this experiment. The rice plants were grown in controlled greenhouses with the temperatures ranging from 20°C at night to a maximum of 36 °C during the day. The inoculation protocols described by Fu et al. [23] and Hu et al [24](2014)were used with minor modifications. At the seventh to eighth stage of panicle development, a mixture of hyphae and conidia (GY900 and PXD25 strains) was injected into each rice panicle. The controls were injected with PS for all experiments. Samples of pathogens with before (mycelium) and after infection were collected, immediately frozen in liquid nitrogen, and kept at −80°C for later use. The samples of strongly virulent strains were named S0 and S1 and weakly virulent strains were named W0 and W1 for before and after infection

What does the abbreviation PS mean?

Answer: PS is the abbreviation of potato sucrose, which has been marked in Materials and methods.

What was the humidity?

This is an important indicator for rice infection. What was the approximate concentration of inoculum

Answer: At the seventh to eighth stage of panicle development, the conidial suspension with a concentration of 3×106 mL-1 was injected into each rice panicle. After inoculation, all the rice plants maintained at 25/30 °C (night/day), covered with an inner solar-shade screen, and automatically sprayed water every 4 h for 20 min to maintain the environment at relative humidity (RH) > 85% for 4 days.

Line 219-221

Figure 3

Needs to improve the quality of the figure. The purple color makes it difficult to see the signature. The font should be enlarged

Answer: According to your suggestions, we have made the corresponding modification in revised manuscript.

We wish the revised manuscript will be accord with your requirements, and thank you for further concerns, comments, and advices.

Many thanks.

Best regards,

Rongtao Fu

18-6-2023

Reviewer 2 Report

I congratulate the authors for their work and manuscript. Manuscript ijms-2443902 "Transcriptomic and metabolomic analyses provide insights into the pathogenic mechanism of the rice false smut pathogen Ustilaginoidea virens" by Fu and collaborators provide a wealth of data investigating this important rice pathogen. The text is clear and should be interesting even to non-specialists of the omics methods. The authors were able to dissect the differences between the strongly and weakly virulent strains, including signaling and metabolic pathways. Expression data for a subset of genes was confirmed by RT-qPCR, strengthening the conclusions of the importance of the highlighted genes. I have only minor suggestions listed below that can be easily modified for the final version.

In the Abstract but also in the main text, I recommend using capital letters in italic for genes.

Line 13: Replace "pathogenic" with "pathogenesis".
Line 21: Replace "in the negative regulator" with "in negative regulation of pathogenesis".
Line 51: Check if the intended term is "antibiotic-resistant" or "fungicide-resistant".
Lines 110 to 115: In this paragraph, clearly state how many replicates for each sample group were analyzed by RNA-seq. Even if the information is repeated in the Statistical Analysis section.
In Table 1, consider using the term "RT-qPCR" instead of "Tubulin" in the header of the last column.
Figure 1 is mentioned but not shown.
Line 242-243: "Signal transduction genes of plant pathogenic fungi..."
Figure 3: Make sure to use high resolution figure in the final version since the text in the figure is very small, so the reader will want to zoom in. Figure 7 as well.
Figure 4: The figure would be more informative if all replicate measurements were shown in the heatmap, and not just one column per treatment. The colors can be based on Z-score that takes into consideration the expression value for the gene across all samples. More similar to Figure 6A.

English is fine overall, but some corrections would improve the quality of the manuscript. Such as:

Line 13: Replace "pathogenic" with "pathogenesis".
Line 21: Replace "in the negative regulator" with "in negative regulation of pathogenesis".
Line 51: Check if the intended term is "antibiotic-resistant" or "fungicide-resistant".
Lines 110 to 115: In this paragraph, clearly state how many replicates for each sample group were analyzed by RNA-seq. Even if the information is repeated in the Statistical Analysis section.
In Table 1, consider using the term "RT-qPCR" instead of "Tubulin" in the header of the last column.
Figure 1 is mentioned but not shown.
Line 242-243: "Signal transduction genes of plant pathogenic fungi..."

Author Response

Dear reviewer

We would like to thank you for giving us the opportunity to revise our manuscript entitled “Transcriptomic and metabolomic analyses provide insights into the pathogenic mechanism of the rice false smut pathogen Ustilaginoidea virens”. We have been carefully revised in the revised manuscript and the every checking was marked in red.

We thank you for their careful read and thoughtful comments on previous manuscript. The point-by-point answers to the comments and suggestions were listed as below.

# Reviewer 2

Comments and Suggestions for Authors

I congratulate the authors for their work and manuscript. Manuscript ijms-2443902 "Transcriptomic and metabolomic analyses provide insights into the pathogenic mechanism of the rice false smut pathogen Ustilaginoidea virens" by Fu and collaborators provide a wealth of data investigating this important rice pathogen. The text is clear and should be interesting even to non-specialists of the omics methods. The authors were able to dissect the differences between the strongly and weakly virulent strains, including signaling and metabolic pathways. Expression data for a subset of genes was confirmed by RT-qPCR, strengthening the conclusions of the importance of the highlighted genes. I have only minor suggestions listed below that can be easily modified for the final version.

In the Abstract but also in the main text, I recommend using capital letters in italic for genes.

Answer: According to your comments, we have used capital letters in italic for genes in revised manuscript.

Line 13: Replace "pathogenic" with "pathogenesis".

Line 21: Replace "in the negative regulator" with "in negative regulation of pathogenesis".

Line 51: Check if the intended term is "antibiotic-resistant" or "fungicide-resistant".

Lines 110 to 115: In this paragraph, clearly state how many replicates for each sample group were analyzed by RNA-seq. Even if the information is repeated in the Statistical Analysis section.

In Table 1, consider using the term "RT-qPCR" instead of "Tubulin" in the header of the last column.

Answer: According to your suggestions, we have made the corresponding modification in revised manuscript.

Figure 1 is mentioned but not shown.

Answer: Figure 1 is presented in the Materials and Methods section

Line 242-243: "Signal transduction genes of plant pathogenic fungi..."

Answer: According to your comments, we have made the corresponding modification in revised manuscript.

Figure 3: Make sure to use high resolution figure in the final version since the text in the figure is very small, so the reader will want to zoom in. Figure 7 as well.

Figure 4: The figure would be more informative if all replicate measurements were shown in the heatmap, and not just one column per treatment. The colors can be based on Z-score that takes into consideration the expression value for the gene across all samples. More similar to Figure 6A.

Answer: According to your suggestions, we have made the corresponding changes to Fig.3, Fig.4 and Fig.7 in the revised manuscript.

Comments on the Quality of English Language

English is fine overall, but some corrections would improve the quality of the manuscript. Such as:

Line 13: Replace "pathogenic" with "pathogenesis".

Line 21: Replace "in the negative regulator" with "in negative regulation of pathogenesis".

Line 51: Check if the intended term is "antibiotic-resistant" or "fungicide-resistant".

Lines 110 to 115: In this paragraph, clearly state how many replicates for each sample group were analyzed by RNA-seq. Even if the information is repeated in the Statistical Analysis section.

In Table 1, consider using the term "RT-qPCR" instead of "Tubulin" in the header of the last column.

Figure 1 is mentioned but not shown.

Line 242-243: "Signal transduction genes of plant pathogenic fungi..."

Answer: According to your suggestions, we have made the corresponding modification in revised manuscript.

We wish the revised manuscript will be accord with your requirements, and thank you for further concerns, comments, and advices.

Many thanks.

Best regards,

Rongtao Fu

18-6-2023